# Strategies for Engineering of Extracellular Vesicles

**DOI:** 10.3390/ijms241713247

**Published:** 2023-08-26

**Authors:** Anna A. Danilushkina, Charles C. Emene, Nicolai A. Barlev, Marina O. Gomzikova

**Affiliations:** 1Laboratory of Intercellular Communications, Institute of Fundamental Medicine and Biology, Kazan Federal University, 420021 Kazan, Russia; 2Laboratory of Molecular Immunology, Institute of Fundamental Medicine and Biology, Kazan Federal University, 420008 Kazan, Russia; 3Department of Biomedicine, Nazarbayev University School of Medicine, Astana 001000, Kazakhstan

**Keywords:** extracellular vesicles, drug delivery, active loading, surface modification, glycan modification, peptides, click chemistry, EVs functionalization, ultrasound, extrusion, freeze-thawing, electroporation, permeabilization

## Abstract

Extracellular vesicles (EVs) are membrane vesicles released by cells into the extracellular space. EVs mediate cell-to-cell communication through local and systemic transportation of biomolecules such as DNA, RNA, transcription factors, cytokines, chemokines, enzymes, lipids, and organelles within the human body. EVs gained a particular interest from cancer biology scientists because of their role in the modulation of the tumor microenvironment through delivering bioactive molecules. In this respect, EVs represent an attractive therapeutic target and a means for drug delivery. The advantages of EVs include their biocompatibility, small size, and low immunogenicity. However, there are several limitations that restrict the widespread use of EVs in therapy, namely, their low specificity and payload capacity. Thus, in order to enhance the therapeutic efficacy and delivery specificity, the surface and composition of extracellular vesicles should be modified accordingly. In this review, we describe various approaches to engineering EVs, and further discuss their advantages and disadvantages to promote the application of EVs in clinical practice.

## 1. Introduction

The ability to release vesicles is an evolutionarily conserved characteristic of Gram-negative and Gram-positive bacteria [1], plants [2], fungi [3], archaea [4], and animals [5] for short- and long-distance communications with the surrounding environment.

Mammalian extracellular vesicles (EVs) are membranous vesicles that are 40–5000 nm in diameter and surrounded by a lipid membrane [6]. They are divided into three main classes: exosomes, ectosomes (or microvesicles), and apoptotic bodies [7]. High biological activity was found in the first two subtypes of extracellular vesicles: microvesicles (50–1000 nm), which are formed due to outward protrusion of the cytoplasmic membrane; and exosomes (30–200 nm), which are formed endogenously from multivesicular bodies [8].

Mammalian EVs serve as a medium for the transfer of information between cells [9]. This can be carried out through the delivery of a large assortment of encapsulated biomolecules—RNA, microRNA, miRNA, DNA, proteins, enzymes, and lipids. Furthermore, EVs are also capable of delivering organelles such as mitochondria, ribosomes, and proteasomes [10]. Information transfer via EVs plays an important role in the process of antigen presentation [11], epithelium–mesenchyme crosstalk in organogenesis [12], signal transmission between pre- and post-synapses [13], and restoration of damaged tissue [14]. The amount and molecular composition of EVs isolated from human biological fluids can be used to diagnose various diseases including cancer [15], Alzheimer’s disease [16], liver [17], lung [18], and diabetic nephropathy [19]. Detailed overviews of the methods used to isolate and characterize extracellular vesicles have been widely described and can be found in reputable reviews [20,21]. Additionally, it should be noted that while working with EVs, there is a list of minimal required information for the studies of extracellular vesicles (MISEVs), as well as protocols suggested by the International Society for Extracellular Vesicles (ISEVs). These cover EV separation/isolation, characterization, and functional studies [22].

EVs produced by human cells are considered potential vehicles for the delivery of drugs and biomolecules [23]. The most promising source of EVs is stem cells, as they have low immunogenicity and high biological activity [24]. Various methods of modifying EVs [25] and loading them with drugs [26] are currently being developed to improve their vector and therapeutic properties. For example, in one study the surface of EVs was functionalized with targeting peptides, which increased the specificity of delivery of several therapeutic molecules. Consequently, this augmented the precision of delivery and increased the efficiency of therapy, thereby reducing the side effects [27]. In another study, superparamagnetic iron oxide nanoparticles (SPIONs) were loaded inside EVs and an external magnetic field was used to direct them to their target [28]. Furthermore, modifications of the glycan surface of glioblastoma-derived EVs resulted in a four-fold increase of EVs internalization by dendritic cells, which can be used to create an EV-based cancer vaccine [29].

This review is focused on the currently existing EVs engineering strategies, which can be roughly divided into two directions: (1) modification of the surface of the EV to enhance its targeting specificity, and (2) modification of the EV content to enhance its biological activity (Figure 1). Accordingly, these two tasks are carried out using two different approaches: cell surface modification is executed at the level of the EV itself and the content of EVs is altered at the level of donor cells (Figure 1). It should be noted, however, that a combination of these methods is also possible, i.e., simultaneous modification of the EVs’ surface and content.

## 2. EV Surface Modification to Enhance Targeting

There are two levels of EV surface modification: modification of the donor cells, followed by the isolation of modified EVs from these cells; and direct functionalization of the EV surface followed by the purification of EVs from the modifying agents, which are described below.

### 2.1. Donor Cell Modification

#### 2.1.1. Genetic Engineering of Donor Cells

The specific markers of EVs are transmembrane proteins of Lamp; glycosylphosphatidylinositol (GPI)-anchored proteins (lipid raft-associated proteins); tetraspanins CD63, CD9, and CD81; and tetraspanin-associated protein CD47, which can be fused with targeting ligands to enhance the targeted EV delivery [30]. To obtain surface-modified EVs, gene modification of donor cells is carried out using plasmid vectors encoding the targeting ligand fused with one of the above transmembrane proteins (Figure 2).

Peptides are the most widely studied targeting ligands due to their small size, high binding affinity, specificity for target cells/tissues, and ability to maintain low immunogenicity and toxicity levels [31,32].

For example, Alvarez-Erviti et al. modified mouse dendritic cells (DCs) to express Lamp2b, an EV membrane protein, which was fused to the neuron-specific RVG (rabies virus glycoprotein) peptide. Modified RVG-EVs effectively delivered their content specifically to neurons, microglia, and oligodendrocytes in the brain after intravenous injection [33].

The targeting of EVs to tumor cells can also be achieved through fusion of the EV membrane protein (Lamp2b) to the integrin-specific peptide iRGD (amino acid sequence: CRGDKGPDC). To achieve this, the authors modified immature DCs by transfecting cells with the pEGFP-C1-iRGD-Lamp-2b plasmid. EVs derived from modified DCs were isolated and additionally loaded with doxorubicin via electroporation. Intravenously injected modified EVs delivered Dox specifically to tumor tissues, which led to the inhibition of tumor growth without side effects on normal cells [34].

Kim et al. transfected HEK293 cells with a vector encoding cardiac-targeting peptide (CTP)-Lamp2b to generate cardiac-targeting EVs. The expression of CTP-Lamp2b on the membrane increased the accumulation of modified EVs in the cells and tissue of the heart, which can be used to treat heart diseases [35].

Li et al. transfected HEK293T cells with plasmid DNA encoding CD9 human antigen R (HuR), which is an RNA-binding protein that interacts with miR-155 with a relatively high affinity. The authors showed that miR-155 was successfully enriched in modified EVs, which is a novel strategy for enhanced RNA cargo encapsulation into engineered exosomes [36].

To increase the selectivity of the EV-based system delivery of protein cargoes to antigen-presenting cells (APCs), the EV-producing cells were modified with pCMV-α-CD206_VSV-G. The authors demonstrated that the attachment of a recombinant llama nanobody α-CD206 to the N-terminus of the truncated VSV-G glycoprotein increased the selectivity of EV cargo delivery, mainly to APCs. This strategy is supposed to be used in the development of drugs for the correction of the immune response in patients with autoimmune, viral, and oncological diseases [37].

Matsumoto et al. also used a genetic engineering approach to produce labeled 125I EVs. The authors modified cells with pCMV–SAV–LA (SAV—streptavidin, LA—lactadherin), which was followed by the isolation of SAV-LA-modified EVs from the culture supernatant. The modified EVs were then incubated with 125I-labeled biotin derivative ((3-125I-iodobenzoyl)norbiotinamide). Radiolabeled 125I-SAV-LA-Exos were administered intravenously to mice and their biodistribution and the effect of cancer-cell derived EVs on tumor growth were evaluated [38].

One of the common features of tumor cells is the high expression of epidermal growth factor receptor (EGFR) on their surface to promote proliferation. The cell-surface-expressed EGFR can be exploited for specific targeting of cancer cells with EVs. One strategy of EV engineering via donor cell modification would be the transfection of cells with vectors encoding for EGFR nanobodies. The latter would target ligands for tumor cells and be fused to GPI (glycosylphosphatidylinositol)-anchored proteins, which were found to be highly enriched on the EVs surface [39]. The surface of modified EVs was enriched in GPI-bound anti-EGFR nanobodies compared to parental cells, which significantly improved the binding of EVs to EGFR-expressing tumor cells [39]. Another example of EGFR-targeted therapy is the modification of donor cells with a vector encoding the GE11 peptide (YHWYGYTPQNVI), which shows high affinity for EGFR-overexpressing cancer cells. Ohno et al. produced EVs modified with the GE11 peptide or EGF to target EGFR-expressing cancer tissues. Intravenously injected EVs delivered let-7a miRNA specifically to xenograft breast cancer cells in RAG2-/- knock-out mice. These data indicated that EVs targeted to EGFR-expressing cells may provide a platform for miRNA replacement therapies in the treatment of various cancers [40].

Yang et al. suggested that a target peptide can be displayed on the EV surface via fusion with CD47, a transmembrane protein abundant on the surface of EVs. In this respect, the cancer cell-specific peptides CDX (FKESWREARGTRIERG) and CREKA (Cys-Arg-Glu-Lys-Ala) were introduced to the N-terminal of CD47, and parent cells were transfected with a plasmid coding for the peptide–CD47 fusion to produce modified EVs. Importantly, the surface-modified EVs showed a significant accumulation in brain tumor in vivo [41].

The genetic engineering approach has a significant drawback, which is that in the process of EV formation, the targeting peptide may be degraded by endosomal proteases inside the producing cell. To protect the targeting peptide from degradation, Hung et al. modified the RVG peptide by adding a glycosylation motif (GNSTM) to it. The authors found that glycosylation protected the peptide from degradation and led to enhanced targeted delivery of EVs to neuroblastoma cells, demonstrating that such glycosylation does not negatively affect the peptide–target interactions [42].

The advantage of the method of genetic engineering of donor cells is the absence of chemical agents that contaminate the final EV product and the high specificity of the modification. The disadvantages of the method include the considerable effort required to create a genetically modified cell line producing the target molecules and the possible impact of the peptide fusion with the targeting part on functions of the EV membrane proteins. Moreover, the genetic engineering method is limited to protein or peptide conjugation, meaning it cannot be used in new approaches using the targeting of EVs, such as those based on aptamers [43,44]. Therefore, there is a need to develop methods for the direct modification of EVs, which are discussed in the next chapter.

#### 2.1.2. Metabolic Engineering of Donor Cells

To target EVs, donor cells can be metabolically altered using endogenous synthesis or specific cleavage processes. To target dendritic cell receptors, high-mannose expressing EVs were developed (Figure 3). High-mannose glycans are recognized by the dendritic cell receptor DC-SIGN (dendritic cell-specific intercellular adhesion molecule-3-grabbing non-integrin), which ensures the targeted delivery of EVs and their contents. Accordingly, donor cells cultured in the presence of kifunensine (a mannosidase inhibitor) accumulated mannose residues in glycoproteins expressed on their cell surface. High-mannose expressing EVs have been isolated from tumor cells to present the tumor-associated antigens to dendritic cells with subsequent priming of tumor-specific T cells. This approach was used to enhance the action of EV-based cancer vaccines [45].

Some disadvantages of the metabolic modification method of donor cells to obtain modified EVs are the limited number of applications and impossibility to extend this approach on isolated EVs or EVs from body fluids [46,47].

### 2.2. Direct EV Functionalization

#### 2.2.1. Peptides

Peptides are characterized by high specificity and affinity, and are the most commonly used tools for targeting cells and EVs [48,49,50]. Through rational design and screening, it is possible to create peptides that target a specific cell protein [51,52]. The modification of EVs is carried out using direct incubation of EVs with peptides that bind non-covalently to the surface of the EV (Figure 4) [53].

The most well-known and widely used peptide is the Arg–Gly–Asp (RGD) peptide with the arginine–glycine–aspartic acid motif. The RGD peptide shows a strong affinity for integrins, especially αvβ3, which is overexpressed on both the endothelial cells and tumor cells [54]. To date, several varieties of the RGD peptide have been created, for example, RGD-4C (Cys–Phe–Cys–Asp–Gly–Arg–Cys–Asp–Cys), cyclic iRGD (CRGDRGPDC), cRGDyK (cyclo(ArgGly–Asp–D–Tyr–Lys)), cRGDfC (cyclo(Arg–Gly–Asp–D–Phe–Cys)), cRGDfK (cyclo(Arg–Gly–Asp–D–Phe–Lys)) and cN–Me–VRGDf (cyclo(MeVal–Arg–Gly–Asp–D–Phe)) [55]. Another type of peptide, the iNGR peptide with an Asn–Gly–Arg motif, also targets integrin-expressing blood vessel cells and tumor cells, and is used as an alternative to RGD peptides.

Gao et al. identified and selected peptide CP05 as it is able to bind to the EV surface protein CD63. The authors further modified the surface of EVs using the CP05 peptide, so that they carried phosphorodiamidate morpholino oligomer (PMO) on their surface, an FDA approved drug for treating Duchenne muscular dystrophy. The data indicated that an intramuscular injection of modified EVs (EVs–CP05–PMO) increased dystrophin protein levels 18-fold, which proved that EVs are an efficient delivery vehicle for PMO [48]. EVs with peptide CP05 conjugates have also been successfully used in the treatment of a number of other diseases, including tumor immunotherapy [56], proliferative retinopathy [57], and traumatic optic neuropathy [58]. CP05 also combines synergistically with other target peptides [59].

Peptides can also be inserted into the surface of EVs using the transfection of parental cells as described above [60], or they can be covalently fused with EV surface proteins using a ligase (enzymatic method). Pham et al. used protein ligase to modify the EV surface. EVs were conjugated with EGFR (epidermal growth factor receptor)-targeting peptide to facilitate the specific uptake of EVs by EGFR-positive cells. In addition, EVs were conjugated with α-EGFR, α-mCherry, and α-HER2 nanobodies to facilitate the specific uptake of EVs by target cells expressing the corresponding receptors. The authors demonstrated that the targeted delivery of EV-encapsulated paclitaxel was significantly enhanced in an EGFR-positive lung cancer xenografted mouse model [27].

Direct EV functionalization with peptides has a number of advantages over other modification methods: (1) the absence of a viral cell transduction step, which reduces the cost and time of production of modified EVs; (2) no risk of oncotransformation of cells due to viral insertion; and (3) peptides and enzymes can be produced in large volumes at low cost.

One of the disadvantages of using peptides as a targeting tool is their unstable nature, i.e., susceptibility to degradation and/or hydrolysis [61]. One possible solution to this problem is the inclusion of D-isomers, which confer greater resistance to proteases [62] or greater stability after the incorporation of glycosylation motifs into the peptides [42].

#### 2.2.2. Glycan Modification

The composition of glycans on the surface of EVs plays an important role in cellular recognition, uptake, and protein binding. Therefore, the manipulation of glycans significantly affects the interaction of EVs with recipient cells and, consequently, their biodistribution [63].

Royo et al. used the enzyme neuraminidase to cleave terminal sialic acid residues from glycoproteins exposed on the surface of EVs (Figure 5) [64]. Compared to native EVs, neuraminidase-treated EVs predominantly accumulated in the axillary lymph nodes and lungs, suggesting their potential use for delivering therapeutic agents to the lymphatic system. The authors suggested that the removal of terminal sialic acid improved the dynamics of EVs by reducing the charge on their surface. This led to an increase in the mobility of EVs due to increased diffusion, improved interaction and uptake by cells. However, from a practical point of view, scaling up deglycosylation to clinical scales may not be feasible given the huge amounts of enzyme that would be required to cleave the abundant sialic acid residues on the surface of EVs.

Another strategy has been proposed by Dusoswa et al. The authors suggested to target tumor EVs to dendritic cells (DCs) using Lewis antigens in order to pulse DCs for the initiation of an anti-tumor immune response. DCs express dendritic cell-specific intercellular adhesion molecule-3-grabbing non-integrin (DC-SIGN, CD209) on their surface, which could bind with high-affinity to the LewisY antigen. The authors prepared LeY-glycolipid, which was inserted into the EVs derived from glioblastoma cells using direct co-incubation. It was found that the EVs modified with LewisY antigen demonstrated a four-fold increase of uptake by monocyte-derived DCs [29].

#### 2.2.3. Click Chemistry

The chemical modification of the surface of EVs occurs by covalently linking the targeting agent to the amino groups of the surface proteins of EVs. The most popular reactions are azide–alkyne cycloadditions: copper-catalyzed azide–alkyne cycloadditions (CuAAC) and strain-promoted alkyne–azide cycloadditions (SPAAC) [65], which belong to click reactions. The difference between the two reactions is that CuAAC can achieve higher rate constants and the simple alkynes needed in these reactions are chemically more accessible and less bulky than the strained cyclooctynes used in SPAAC. However, the latter approach does not require the addition of Cu(I) ions, which cause oxidative damage to the surface proteins of EVs and are toxic to cells.

An example of a CuAAC modification is the work of Smyth et al. The authors have developed a method for modifying EVs that consists of chemical modification of EV proteins with alkyne groups and subsequent conjugation to a model azide, azide-fluor 545 (Figure 6) [66]. This conjugation did not affect the size and binding of EVs to cells; in other words, it did not change the structure or function of EVs.

Jia et al. also modified the surface of EVs using the CuAAC click chemistry reaction to obtain glioma-targeting EVs. In the first stage, the authors performed a conjugation of the alkyne group with the protein (the phosphatidylethanolamine) on the EVs’ membrane. In the second stage, RGE-peptide (neuropilin-1-targeted peptide (RGERPPR, RGE)) was conjugated to the alkyne group on the EVs’ membrane [67].

An example of the application of a SPAAC reaction is the modification of EVs with a cyclic cRGD peptide that has a high binding capacity to the αvβ3 integrin overexpressed in endothelial cells during cerebral ischemia. The authors showed that cRGD-modified EVs targeted the lesion region of the ischemic brain after intravenous administration and were able to deliver curcumin loaded in their contents to the inflammatory site [68].

Lee et al. used a combined EV modification method consisting of metabolic cell labeling with strain-promoted azide–alkyne click (SPAAC) chemistry. In the first step, tetra-acetylated N-azidoacetyl-D-mannosamine (Ac4ManNAz) was metabolically incorporated into the sialic acid of cells and EVs glycoconjugates. In the second step, azide-containing EVs were labeled with azadibenzylcyclooctyne (ADIBO)-sulfo-Cy3 or Cy5.5 fluorescent dyes using bioorthogonal click chemistry. Modified fluorescent-labeled EVs were also used to assess cellular uptake and the in vivo tracking [69].

Another example of a SPAAC reaction is in the work of Wang et al. The authors also applied a modification of EVs at two levels—metabolic engineering of donor cells and modification of EVs. Donor cells (murine melanoma cell line B16F10) were cultured in the presence of L-azidohomoalanine (AHA) or tetraacetylated N-azidoacetyl-Dmannosamine (ManNAz) for three days. AHA is an azide-bearing amino acid analogue of methionine, which is introduced into newly synthesized proteins of donor cells, and hence also into EVs. Similarly, when cells were cultured in the presence of ManNAz–azide-bearing saccharides, azide groups were incorporated into the surface glycans of donor cells and EVs. In the next stage, the authors isolated modified EVs, bearing azide groups on their surface. Subsequently, they performed click conjugation on the modified EVs using the reaction between an azide and dibenzobicyclooctyne (DBCO). DBCO-Cy3 was added to the modified EV suspension to fluorescently stain the EVs. An intense fluorescence of the modified EVs confirmed the incorporation of azides into the EVs, which indicated the effectiveness of the method. This approach can be further used for extended modification of EVs and their use as a drug delivery platform [70].

Another hydrazine–aldehyde-based click reaction strategy was used by Zhang et al., who used a reaction between 6-hydrazinonicotinate acetone hydrazone (HyNic) with 4-formylbenzoate (4FB) to covalently attach quantum dots (QDs) to the surface of EVs. This method allowed the visualization of EV trafficking and function in vitro and in vivo. QD-labeled EVs showed better resistance to photobleaching than EVs labeled with the widely used DiI membrane dye [71].

#### 2.2.4. Sulfhydryl–Maleimide Crosslinking

Sulfhydryl is widely distributed in most proteins, including membrane proteins. Therefore, EVs can be labeled with various functional molecules via the biocompatible interlinkage between sulfhydryl and maleimide. Fan et al. described a DNA hinge-based labeling strategy of EVs based on sulfhydryl–maleimide crosslinking. During the first step, the authors engineered a DNA hinge (functionalized DNA, maleimide-5′-AAAAAA-3′-biotin) to attach it to the surface of EVs via a sulfhydryl–maleimide reaction. In the second step, the authors added streptavidin labeled quantum dots (QDs), which can recognize and bind to biotin of the DNA hinge, i.e., EV–maleimide-5′–AAAAAA-3′-biotin (Figure 7). By doing this, the authors obtained modified EV–DNA–QDs with anchored QDs on the surface of EVs, which can be applied as a specific agent for tumor labeling [72].

Summarizing the above, it can be concluded that the chemical modification of EVs has great potential for large-scale implementation due to its ease of use and the wide range of targeting agents. However, the main disadvantage of the chemical modification of EVs is that it is a cumbersome multi-step modification process with multiple purification steps that in the loss of EVs. In addition, it is necessary to comprehensively characterize and control the quality of the obtained EVs, since modifications can affect their properties, natural activity, and biodistribution.

## 3. EV Content Modification to Enhance the Activity

### 3.1. Donor Cell Modification

#### 3.1.1. Genetic Engineering of Donor Cells

The extent of the therapeutic effect of EVs is largely dependent on their content, which can be modified in order to stimulate or inhibit certain processes in target cells. For example, mouse macrophages, which were used as donor cells for EV isolation, were transfected with plasmid DNA encoding glial-cell-line-derived neurotrophic factor (GDNF) to enhance the neuroprotective activity of EVs. As expected, it was found that EVs collected from the conditioned media of GDNF-transfected macrophages contained GDNF proteins as well as genetic material—GDNF-encoding DNA. Moreover, applying GDNF–EVs via intranasal administration to the mouse model of Parkinson disease (transgenic Parkin Q311(X)A mice) led to a significant improvement in mobility, an increase in neuronal survival, and a decrease in neuroinflammation in mice [73].

Li et al. genetically engineered T-cell-derived EVs, which exhibited PD-1 receptor of programmed cell death ligand 1 (PD-L1) on the surface to enhance their anti-tumor activity. Donor cytotoxic T lymphocyte cells (CTLL-2 line) were transduced with lentiviruses encoding PD-1. EVs expressing PD-1 (PD-1 EVs) neutralized PD-L1 and effectively reinvigorated the activity and proliferation capacity of CD8+ effector T cells. Moreover, PD-1-containing EVs also directly attacked tumor cells via Fas-ligand (FasL) and granzyme B (GzmB) [74].

To ameliorate ischemia/reperfusion (I/R)-injured endothelial cell (EC) function, Pan et al. produced EVs containing microRNA-126 (miR-126). miR-126 is an important regulator of endothelial cell (EC) function and angiogenesis. To load EVs with miR-126 the authors transfected mesenchymal stem cells with Lv-miR-126 and isolated miR-126-loaded EVs. Treatment of I/R-injured ECs with the engineered EVs enhanced the survival and angiogenic function of injured ECs, decreased the expression of caspase-3, and increased the expression of angiogenic and growth factors [75].

As a new therapeutic strategy for neurodegenerative diseases, it was proposed to use EVs derived from genetically modified macrophages. Donor macrophages were transfected with a plasmid DNA encoding for an antioxidant enzyme catalase. As a result, isolated EVs were packed with the corresponding pDNA and mRNA, an active catalase itself, and transcription factor NF-kb. The authors demonstrated that EVs efficiently transferred their contents to neurons in vitro [76].

To treat morphine addiction, Liu et al. applied two strategies of EV modification—EV content modification and EV surface modification. Donor HEK 293T cells were co-transfected with an opioid receptor mu (MOR)-specific small interfering RNA (siRNA) and the RVG-Lamp2b plasmid. RVG–EVs efficiently delivered siRNA to the central nervous system and downregulated MOR expression levels [40]. Bellavia et al. also applied a double EV modification by transfecting HEK293T cells with siRNA, targeting the BCR–ABL oncogene and a plasmid encoding a fusion protein (IL3-Lamp2b). As a result, EVs expressed IL3-fused Lamp2b proteins on their surface and exhibited an increased binding affinity to IL3 receptor-expressing cells—Chronic Myeloid Leukemia (CML) blasts. Modified EVs loaded with imatinib or BCR–ABL siRNA showed a notable ability for reducing tumor proliferation in vivo and in vitro [77].

It is well known that the volume of EV secretion is cell line-dependent. Thus, the genetic makeup of the prospective cell line should be examined before choosing it as a source of EVs. For example, it is known that one of the main tumor suppressors, p53, controls EV secretion. P53 operates chiefly as a transcription factor and is tightly controlled by various ubiquitin ligases at the protein level [78]. As part of the DNA damage response, p53 transactivates the tumor-suppressor-activated pathway-6 (TSAP6) gene, whose product promotes exosomal secretion, in addition to its role in the regulation of cell-cycle arrest and apoptosis [79]. Importantly, in addition to the p53-dependent secretion of EVs, TSAP6 can also facilitate the production of exosomes independent of p53 status [80]. Thus, the forced overexpression of TSAP6 may positively affect the efficacy of EV production in the target cell line. Alternatively, inhibitors of Mdm2, the principle E3 ligase of p53 [81,82,83], or small molecules that stabilize p53 [84,85] can be applied prior to the secretion of EVs.

#### 3.1.2. Direct Co-Culture of Donor Cells with Drugs

Cells can take up a drug solution or SPIONs via endocytosis when co-incubated. Donor mouse MSCs of the SR4987 line (bone marrow-derived mesenchymal stromal cell line isolated from BDF/1 mice) were loaded with Paclitaxel (PTX) by exposing cells in vitro to a high concentration of the drug. Cells were incubated in PTX solution for 24 h, followed by washing and culturing for 48 h, to produce EVs. The authors observed that the PTX treatment did not impair EV formation. The EVs secreted by SR4987-PTX were loaded with PTX and demonstrated a significant anti-proliferative activity against human pancreatic adenocarcinoma cells [86].

A similar result was obtained for HepG2 cells incubated with various anti-tumor agents: PTX, Etoposide, Carboplatin, Irinotecan, Epirubicin, and Mitoxantrone. EVs released from drug-treated HepG2 cells showed strong anti-proliferative activity against the human pancreatic cell line CFPAC-1 and induced immunogenicity and HSP (heat shock proteins)-specific NK cell responses [87].

Tumor cell-derived EVs were suggested to be used as vectors to deliver chemotherapeutic drugs. To load EVs with drugs, donor cells (tumor cell line H22 or A2780) were incubated with methotrexate (MTX) or doxorubicin; then, to induce EV release, donor cells were irradiated with ultraviolet light was used for apoptosis induction. It was found that drugs were packaged into the released EVs, which in turn induced tumor cell death. MTX-encapsulating EVs were used for anti-cancer therapy in vivo and led to the inhibition of tumor growth without adverse effects [88]. A similar method of EV loading and production was applied to evaluate the anti-tumor activity of autologous MTX-loaded EVs in a pilot human study. Patients with advanced lung cancer and malignant pleural effusion (MPE) received a single dose of MTX–EVs via intrapleural infusion. It was found that manufacturing and infusing MTX–EVs were safe, without toxic effects, and that infusion of MTX–EVs led to notable reductions in the number of tumor cells and CD163+ macrophages in MPE, as well as to the stimulation of IL-2 and IFN-γ release, which can activate cytotoxic T lymphocyte (CTL)/T helper 1 (TH1) responses and elicit anti-tumor immunity [89].

EVs can be loaded not only with drugs, but also with nanoparticles. MSCs isolated from human umbilical cord tissue were incubated with Fe3O4 nanoparticles (NPs) for 24 h with subsequent EV isolation from a conditioned medium. It was shown that an in vivo systemic injection of NPs containing EVs (EVs–NPs) with magnetic guidance significantly increased the number of EVs–NPs at the injury site (cutaneous wounds), where EVs–NPs enhanced endothelial cell proliferation, migration, and angiogenic tubule formation [90]. EVs loaded with nanoparticles are used for MRI biodistribution studies as well as to target delivery of EVs contents into cells. However, the efficiency of uptake of nanoparticles by parent cells is low, and the amount of these nanoparticles trapped inside EVs cannot be controlled.

### 3.2. Loading Drugs into EVs

Therapeutics encapsulated within EVs (or any other lipid vesicles) can be delivered using two main approaches: passive or active loading. Passive loading is mediated via incubation of EVs in a drug or dye solution that enters the EVs through diffusion or endocytosis [91]. Using this approach, EVs were successfully loaded with several low molecular weight compounds including antioxidant molecule curcumin [92], anti-cancer drugs, doxorubicin (Dox) [34], and paclitaxel (PTX) [93], to name just a few. Although this approach is simple and easy to implement, it is rather inefficient because of the small volume of EVs, small pore size, and hydrophobic properties of their membranes, which collectively limit the efficiency of drug penetration.

On the other hand, active loading of EVs includes the following methods: ultrasound, extrusion, freeze/thaw cycles, electroporation, and treatment with permeabilizing agents. These approaches are briefly discussed below.

#### 3.2.1. Ultrasound

EVs can be mixed with therapeutic drugs or proteins and then sonicated. Depending on the amount of ultrasound the EVs are exposed to, this may either partially disrupt the integrity of the EV membranes and allow the drug to diffuse inwards, or cause complete membrane disruption, release of the EV contents, and subsequent self-assembly of the membranes to entrap the drug solution in which the sonication was performed.

Haney et al. demonstrated that the EV membrane microviscosity is significantly reduced after sonication [94]. However, the process of ultrasonic deformation of the membrane does not significantly affect the membrane-bound proteins or lipid composition of the EV. It was found that the integrity of the membrane was restored within an hour when EVs were incubated at 37 °C. In some cases, drugs were not only encapsulated inside the EV, but also adhered to the outer layer of the membrane; as a result, two phases of drug release were observed. The first one was the fast release phase, which resulted from the release of the drug attached to the outer layer of the EV, followed by a slow release of the drug encapsulated within the EV [95].

#### 3.2.2. Extrusion

Some studies have developed an approach to create EV-like nanovesicles that are assembled from the plasma membrane of parent cells through a membrane extrusion process. EV-like nanovesicles are mainly produced using extrusion through successive polycarbonate membrane filters with decreasing pore size. Compared to natural extracellular vesicles, these EV-like nanovesicles exhibit similar size, shape, zeta potential, and biomolecules as cell-secreted extracellular vesicles [96]. These EV-like nanovesicles erase the limitations associated with a low yield, hence offering a means for the large-scale production of extracellular vesicles.

EVs are mixed with the drug and the mixture is subsequently loaded into a lipid extruder based on a syringe with a membrane that has a pore size of 100–400 nm. During the extrusion, the EV membrane is ruptured and mixed with a drug. However, Fuhrmann et al. reported that the loading of EVs isolated from MDA-MB231 breast cancer cells with porphyrin using the extrusion method altered the zeta potential of the original EVs, the structure of membrane proteins, and induces cytotoxicity, whereas loading of porphyrin using other methods (such as electroporation, saponin treatment, or hypotonic dialysis) did not elicit cytotoxicity. This may be a result of intense extrusion (EVs were extruded more than 30 times), which transformed the properties of the membrane [97]. In another study, loading of catalase into RAW264.7 macrophage EVs using 10 extrusions did not induce cytotoxicity, but instead, the loaded EVs showed greater neuroprotective activity than EVs prepared using the freeze/thaw cycle or simple incubation methods [94]. Therefore, it is necessary to test the biological activity and safety of EVs modified using this method.

SPION-containing EV-like vesicles were generated using extrusion of human mesenchymal stem cells (hMSC) treated with SPION (hMSC–SPION). The use of magnet-guided navigation caused an accumulation of EV-like nanovesicles administered intravenously in the injured spinal cord [98]. In addition, magnetic EV-like nanovesicles obtained from MSCs containing SPION can be used to treat ischemic stroke [99] and cardiovascular diseases [100].

#### 3.2.3. Freeze and Thaw Cycles

The freeze–thaw procedure for loading EVs consists of mixing drugs with EVs at room temperature followed by cycles of flash freezing at −80 °C or in liquid nitrogen, and subsequent thawing at room temperature. This process is repeated at least three times to ensure drug encapsulation. This method allows the destruction of plasma membranes through the temporary formation of ice crystals [101], which cause significant structural and functional changes, including lateral phase separation of the membrane components and membrane fusion. Therefore, this method may cause EV aggregation, resulting in a wide distribution of EV sizes. The loading efficiency of the freeze–thaw method is generally lower than that of sonication or extrusion methods. The method is used to fuse membranes between EVs and liposomes and create particles that mimic EVs. Sato et al. used a freeze–thaw method to fuse EVs from RAW264.7 macrophages with phospholipid-based liposomes. The number of freeze–thaw cycles was found to affect the dilution ratio of lipids in the resulting EV liposome particles [101].

#### 3.2.4. Electroporation

This is a process in which short electrical impulses create temporary pores in the plasma membrane [102]. During electroporation, drugs or nucleotides are able to diffuse into the EV through these pores. This method is widely used to load relatively large siRNA or miRNA molecules into EVs that cannot spontaneously diffuse into EVs, such as small hydrophobic molecules. Apart from nucleic acids, electroporation is used to load small hydrophilic molecules into EVs, such as TMP (5,10,15,20-tetrakis (1-methyl-4-pyridinio) porphyrin tetra (p-toluenesulfonate)), which is used for photodynamic treatments [97]. Electroporation also allows superparamagnetic iron oxide nanoparticles (SPION) to pass through temporary pores in the EV membrane to create magnetic EVs that are used for in vitro and in vivo imaging using MRI approaches [103].

However, electroporation can cause EV aggregation. Johnsen et al. reported that the addition of the disaccharide trehalose to the electroporation buffer assisted in maintaining structural integrity of EVs and inhibited the aggregation of EVs derived from adipose tissue stem cells [104].

Additionally, there are concerns about the potential impact of electroporation on EV functionality. To this end, Fuhrmann et al. reported that the overall morphology of EVs does not change after electroporation [97]. In another study, it was shown that there were some minor differences in the topography of EVs after loading them with siRNA using electroporation. Such EVs had a slightly larger average diameter and surface potential [105]. Pomatto et al. found no loss of intrinsic EV content due to electroporation. However, the authors showed that, depending on the mode (voltage) of electroporation, there is a risk of damaging the EVs [106]. Thus, further studies are required to accurately assess the potential impact of electroporation on the functionality of extracellular vesicles.

#### 3.2.5. Treatment with Permeabilizing Agents

Saponin is a surfactant molecule that can form complexes with cholesterol in cell membranes to generate pores. This, in turn, leads to an increase in membrane permeability [107]. Saponin increases the permeability of the EV membrane for various proteins including a model catalase enzyme. Importantly, this method did not destroy the structure and enzymatic activity of the catalase complex [108]. It has been shown that hydrophobic compounds are loaded more efficiently into EVs using saponin than the hydrophilic ones [97]. At the same time, saponin treatment helped loading a relatively hydrophilic molecule, porphyrine (porBA), 11 times more efficiently into EVs than the passive loading without using saponin [97]. However, there are concerns about the hemolytic activity of saponin in vivo [107]. Therefore, the concentration of saponin for loading a drug should be carefully titrated, and the EV must be purified after treatment with saponin.

The described approaches used to load molecules and drugs into EVs have varying efficiencies. As mentioned earlier, sonication and extrusion provide the highest loading efficiency inside the EVs compared to freeze–thaw cycles and passive incubation [94]. However, loading efficiency is not the only factor to consider when choosing a method. It is often a priority to preserve the integrity of the EV membrane and cytoplasmic proteins, and to avoid contamination with immunogenic and toxic substances.

The advantages and disadvantages of the reviewed EV modification strategies are summarized in Table 1. The possible applications are also mentioned.

Thus, each of these approaches has its own advantages and disadvantages, which should be considered during the development of a therapeutic strategy based on modified vesicles. In general, modification methods that retain the biocompatibility of EVs and have minimal destructive effects on the molecules and physicochemical parameters of EVs (such as genetic or metabolic engineering of donor cells and EV functionalization with peptides) are more expensive or less effective.

To increase the efficiency of modification and reduce costs, methods of EV functionalization (such as click chemistry and active loading using ultrasound/extrusion/freeze–thaw cycles/electroporation/permeabilization) are being developed, but they show side effects such as oxidative, destructive effects on molecules, or aggregation of EVs. Therefore, modified EVs obtained using any of the described methods must undergo quality control to evaluate their morphology, integrity, homogeneous size, absence of EV aggregates, and absence of contaminants in the form of xenogenic components of the nutrient medium [111].

It is important to note that the size of the obtained EVs and the presence of a tendency to aggregate determine the possible route of administration of these vesicles. Furthermore, each batch of EVs produced must undergo verification of its molecular composition and biological activity [111].

We believe that it is worth choosing a specific strategy for modifying vesicles based on their intended application. For example, giving priority to high encapsulation efficiency and applicability for most cargoes in the case of anti-tumor therapy. On the other hand, for regenerative medicine and immunomodulatory therapy, the most important criteria are the minimal destructive effects on molecules, biocompatibility, and unchanged physicochemical parameters of EVs.

## 4. Challenges and Future Directions in EV Engineering

Since the strategies mentioned above for modifying vesicles do not significantly alter the membrane composition and the array of surface molecules of vesicles, they exhibit a similarity to the native vesicles’ pattern of biodistribution and uptake by recipient cells in the body. The implementation of vesicle targeting strategies (such as genetic engineering of donor cells, metabolic engineering of donor cells, functionalization of EVs with peptides/glycan modifications, click chemistry, and sulfhydryl–maleimide crosslinking) aims to enhance the preferential interaction with target cells.

However, a challenge that persists with modified vesicles is the rapid clearance of EVs after intravenous injection. Studies have demonstrated that EVs are primarily engulfed by macrophages in the liver and spleen, as well as by endothelial cells in the lungs [112]. Therefore, the issue of maintaining EV stability in circulation remains, necessitating further improvement and research.

The next aspect concerns the off-target effects of EVs (interactions with non-target or non-diseased cells). The development of new strategies for achieving more specific targeting of vesicles has the potential to reduce these off-target interactions. However, much remains to be understood about the target-cell specificity of modified EVs and improving tissue-specific targeting [113].

During the formation process, EVs naturally contain a diverse array of bioactive molecules within their composition, which are also present in modified EVs and could potentially lead to unexpected side effects. Therefore, a precise analysis of the EV content, the fusion and uptake processes of the target tissue, the delivery of therapeutic cargo to the target tissue, and the impact on the biological function of the target tissue or cell becomes essential [113].

Additionally, the utilization of EVs in clinical practice is currently challenging for several reasons: (1) cells secrete a limited number of EVs, which is insufficient for clinical translation; (2) existing methods for isolating EVs are either time-consuming or expensive; (3) there is no clinically viable method for the scalable production of EVs with consistent properties; (4) production protocols adhering to Good Manufacturing Practice (GMP) and controlling the “quality” of the produced EVs have not been developed [111]. Therefore, it is crucial to develop a scalable engineering strategy, a scalable EV production procedure, and a scalable EV isolation protocol to achieve reproducibility in EV manufacturing. This includes establishing standardized procedures for EV storage and shipping, defining critical quality attributes of EVs, and providing regulatory guidance. These steps will ensure the safety and efficacy of EV applications [113].

## 5. Conclusions

The aim of this review was to discuss the advantages and disadvantages of strategies for modifying EVs, as well as the possibilities for the clinical use of modified EVs. EVs are now considered a promising delivery vehicle and therapeutic tool that can be adapted for clinical use. The goal of engineering EVs is to improve their targeting specificity and increase the biological payload. To achieve these goals, methods for modifying EVs have been developed, which, in principle, are divided into two approaches modification at the level of donor cells and direct modification of EVs. Each of these approaches has its own advantages and disadvantages, which determine its applicability. For example, genetic modification of donor cells is a more time-consuming and expensive method; however, once a line of producer cells has been created, it is possible to constantly harvest EVs that retain their biologically active content, structural integrity, and functions. In addition, this method does not require the stage of additional purification of EVs. At the same time, a method of direct modification of EVs using peptides, glycan modification, and click chemistry is less time-consuming and labor-intensive, but requires additional purification steps and verification of the physicochemical parameters of EVs before use. In general, the above methods are used for the production of EVs for regenerative medicine and targeted delivery.

When loaded with drugs, the molecular composition of donor cells and EVs changes. In the case of active loading of EVs with drugs, their composition, morphology, charge, colloidal stability, and size can also change. This unwanted effect requires additional stages of purification and verification of the properties of EVs before use. However, the method of loading donor cells or EVs with drugs allows high concentrations of therapeutic substances to be achieved inside EVs. Therefore, these methods are used to create new antitumor therapeutics based on modified EVs.

EVs provide a new prospective tool to modern medicine. The methods considered in this review make it possible to create a wide range of highly specific and effective drugs based on EVs.

## Figures and Tables

**Figure 1 ijms-24-13247-f001:**
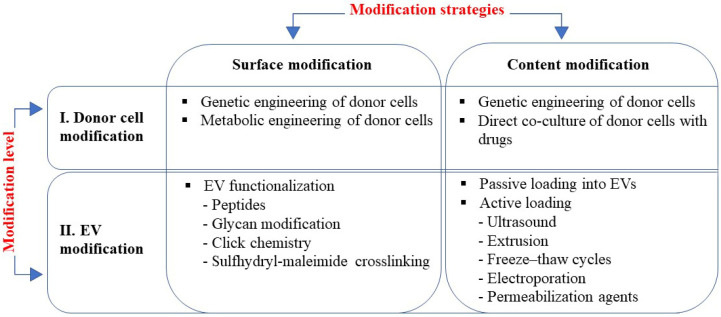
EV engineering strategies rely on two approaches: modification of surface to enhance targeting; and modification of content to enhance activity, which can be applied at two levels of modification, i.e., modification of donor cells and EVs modification. Methods used at each modification strategy are listed in boxes.

**Figure 2 ijms-24-13247-f002:**
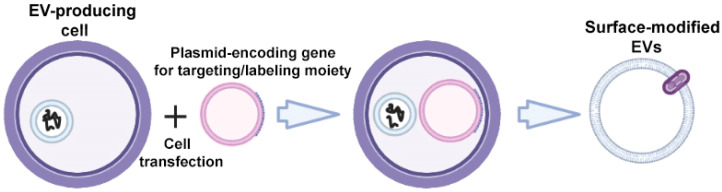
Strategy for surface modification of extracellular vesicles: genetic engineering of donor cells. Partially created with BioRender.com (accessed on 1 May 2023).

**Figure 3 ijms-24-13247-f003:**
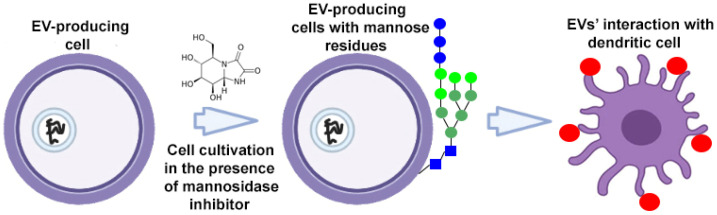
Strategy for surface modification of extracellular vesicles: metabolic engineering of donor cells. Partially created with BioRender.com (accessed on 1 May 2023). 
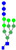
—mannose residue, 
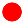
—extracellular vesicle.

**Figure 4 ijms-24-13247-f004:**
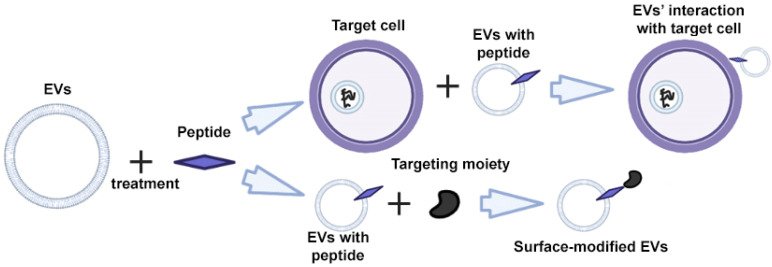
Strategy for surface modification of extracellular vesicles: direct EV functionalization with peptides. Partially created with BioRender.com (accessed on 1 May 2023).

**Figure 5 ijms-24-13247-f005:**
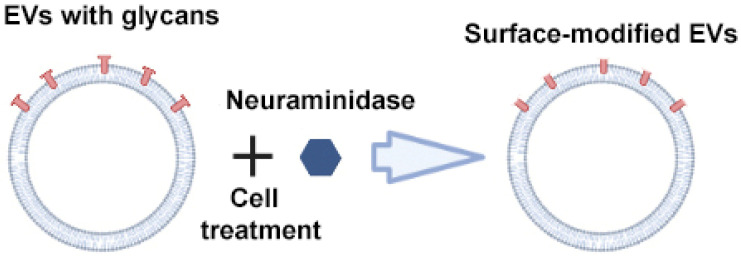
Strategy for surface modification of extracellular vesicles: glycan modification. Partially created with BioRender.com (accessed on 1 May 2023).

**Figure 6 ijms-24-13247-f006:**
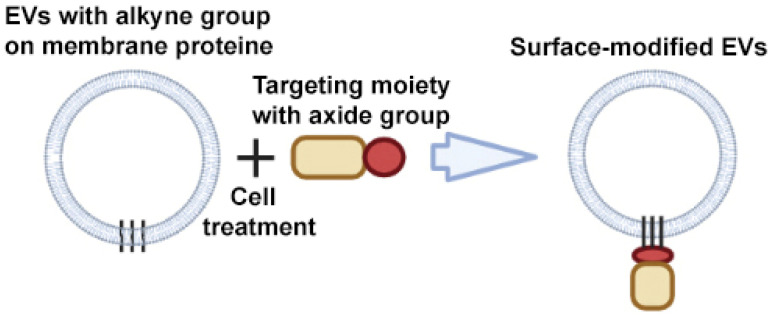
Strategy for surface modification of extracellular vesicles: click chemistry. Partially created with BioRender.com (accessed on 1 May 2023).

**Figure 7 ijms-24-13247-f007:**
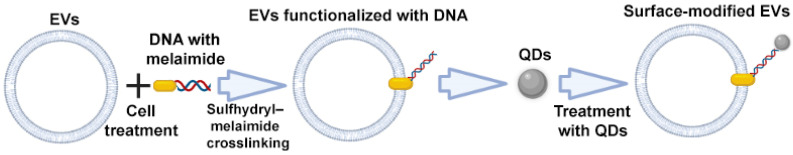
Strategy for surface modification of extracellular vesicles: sulfhydryl–maleimide crosslinking. Partially created with BioRender.com (accessed on 1 May 2023).

**Table 1 ijms-24-13247-t001:** Advantages and disadvantages of EV modification strategies.

Modification Strategy	Advantages	Disadvantages
Genetic engineering of donor cells	Establishing a stable producing cell line; constant harvest of EVs; minimal destructive effect on molecules; biocompatible composition; physicochemical parameters of EVs are not disturbed.	Time-consuming procedure; complex and expensive method of establishment of producing cell line; not applicable to isolated EVs or EVs from body fluids.
Metabolic engineering of donor cells	Simplicity of implementation; constant harvest of EVs; minimal destructive effect on molecules; biocompatible composition; physicochemical parameters of EVs are not disturbed.	Limited number of applications; not applicable to isolated EVs or EVs from body fluids.
EV functionalization: Peptides	Simplicity of implementation; low cost; minimal destructive effect on molecules; biocompatible composition; physicochemical parameters of EVs are not disturbed.	Susceptibility of peptides to degradation.
EV functionalization: Glycan modification	Simplicity of implementation; minimal destructive effect on molecules; biocompatible composition.	Limited number of applications; expensive method; requires verification of the physicochemical parameters of EVs.
EV functionalization: Click chemistry	High selectivity; simplicity of implementation; wide variety of applications.	Requires additional purification (to remove trace Cu); side reactions such as oxidation of amino acids and loss of activity of bioconjugates [109]; requires verification of the physicochemical parameters of EVs.
EV functionalization: Sulfhydryl–maleimide crosslinking	Simplicity of implementation; widely distributed sulfhydryl groups in proteins; minimal destructive effect on molecules; biocompatible composition.	Limited number of applications; requires verification of the physicochemical parameters of EVs.
Passive loading into EVs	Simplicity of implementation; low cost; minimal destructive effect on molecules.	Low encapsulation efficiency; limited efficiency of drug penetration.
Active loading: Ultrasound	Simplicity of implementation; low cost; applicable for most cargoes; high encapsulation efficiency [110].	Destructive effect on molecules; requires verification of the content, morphology, size, and function of EVs; EV aggregation, which complicates intravenous administration.
Active loading: Extrusion	Simplicity of implementation; low cost; applicable for most cargoes; high encapsulation efficiency.	Destructive effect on molecules; cytotoxicity; alters zeta potential of EVs; requires verification of the content, morphology, size, and function of EVs; EV aggregation, which complicates intravenous administration.
Active loading: Freeze–thaw cycles	Simplicity of implementation; low cost; applicable for most cargoes.	Low encapsulation efficiency; EV aggregation, which complicates intravenous administration; requires verification of the content, morphology, size, and function of EVs.
Active loading: Electroporation	Simplicity of implementation; low cost; applicable for most cargoes; high encapsulation efficiency.	EV aggregation, which complicates intravenous administration; alters the physicochemical and morphological characteristics of EVs; requires verification of the content, morphology, size, and function of EVs.
Active loading: Permeabilization agents	Simplicity of implementation; low cost; applicable for most cargoes.	Hemolytic activity of saponin; requires additional step of purification; requires verification of the content, morphology, size, and function of EVs.

## Data Availability

All data generated and analyzed during this study are included in this published article. The data that support the findings of this study are available from the corresponding author upon request.

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
