# Peer review of "Strategies for Engineering of Extracellular Vesicles"

_ijms, 2023, doi:10.3390/ijms241713247_

Round 1

Reviewer 1 Report

It would be best to break up Figure 2 into different figures for each section. This review article would also be improved if in the text or figure, the method by which this EV modifications are made (e.g. electroporation). Moreover, I would like to know about the authors perspective regarding how these modifications would affect EV incorporation and/or processing by recipient cells and effects to non-diseased cells. 

Please do a thorough grammar and spelling check. 

e.g. "This can be carried out by delivering of a large assortment"

I think you meant delivery instead of delivering. 

Author Response

Comment

It would be best to break up Figure 2 into different figures for each section.

Reply

Thank you for your work and careful reading of the manuscript. As it was suggested, we have divided Figure 2 into different figures (Figures 2-7).

Comment

This review article would also be improved if in the text or figure, the method by which this EV modifications are made (e.g. electroporation).

Reply

As it was suggested, we have added short descriptions in the figures about the methods by which EVs modifications were made (please see modified Figures 2-7).

Comment

Moreover, I would like to know about the authors perspective regarding how these modifications would affect EV incorporation and/or processing by recipient cells and effects to non-diseased cells.

Reply

We thought about this circumstance and added discussion to the text of the manuscript (Lines 582-588). Since the reviewed strategies for modifying vesicles do not dramatically change the membrane composition and repertoire of surface molecules of EVs, they show a similar to native vesicles pattern of biodistribution and incorporation by recipient cells in the body. However, when EVs targeting strategies are applied, there is a shift towards preferential interaction with target cells.

In response to a question about effects to non-diseased cells: indeed, despite the development of targeting strategies, non-specific interaction of EVs with non-diseased cells is still a problem. Therefore, we have devoted attention to this issue in a separate chapter (Lines 589-598, highlighted in blue). One of the main problems here is EVs rapid uptake by macrophages in the bloodstream. The next aspect is EVs off-target effects (interaction with non-target or non-diseased cells). We believe that development of new strategies for more specific targeting of vesicles can reduce the off-target interaction of EVs.

Comment

Comments on the Quality of English Language Please do a thorough grammar and spelling check.  e.g. "This can be carried out by delivering of a large assortment" I think you meant delivery instead of delivering.

Reply

Thank you for pointing out this typo. We turn to the help of a native speaker of the English language on the final proofreading of the manuscript.

Reviewer 2 Report

The authors present a comprehensive review on the engineering strategies to achieve EVs for drug delivery applications. The review is well written and various engineering methods are elaborated in detail with appropriate references. That said, I suggest some important additions to the review for publication:

- I believe the review is missing a short section on EV production and purification methods. A section with this information upfront in the review in the form of a table or text will greatly enhance the quality of the review.

- I would suggest the authors present a comprehensive table with different EV engineering methods with their advantages and disadvantages especially for active loads strategies upon which the authors can draw a critical conclusion on best methodologies.

- Further, another table with EVs engineering methods and specific applications they have been used for. For example, the authors can try to draw a trend on whether a particular strategy is used specifically for a particular type of application such as size exclusions is need for in vivo applications whereas it might not be necessary for oral delivery of EVs.

- Finally, section on challenges the community is facing in EV engineering need to be discussed. Critical comments on these will provide basis for new research and help accelerate the field.

- Minor comment: Figure 2 can be moved to page 3 or split into multiple figures and added at their respective sections (optional).

Author Response

Comment

- I believe the review is missing a short section on EV production and purification methods. A section with this information upfront in the review in the form of a table or text will greatly enhance the quality of the review.

Reply

Thank you for your valuable comments and careful reading of the manuscript. We agree that your suggestion would greatly improve the manuscript. We thought about the possibility of expanding the article with the proposed sections, however, we realized that we are limited by the size of the mini review format. Therefore, we suggested and referred to the relevant qualitative reviews about EV production and purification methods (Lines 46-51, highlighted in blue).

Comment

- I would suggest the authors present a comprehensive table with different EV engineering methods with their advantages and disadvantages especially for active loads strategies upon which the authors can draw a critical conclusion on best methodologies.

Reply

As it was suggested we have done a comprehensive table summarizing advantages and disadvantages of reviewed modification strategies (please see Table 1). Additionally, a critical discussion and conclusions on applicability of methods have been made (Lines 558-580, highlighted in blue).

Comment

- Further, another table with EVs engineering methods and specific applications they have been used for. For example, the authors can try to draw a trend on whether a particular strategy is used specifically for a particular type of application such as size exclusions is need for in vivo applications whereas it might not be necessary for oral delivery of EVs.

Reply

As it was suggested we began to build a table with specific applications of modified EVs and realized that they were used or can be used as in regenerative medicine as well in anticancer therapy. So, we pointed out limitation of in vivo applications in the text (in disadvantages in Table 1) and added the following text in the manuscript (highlighted in blue):

« To increase the efficiency of modification and reduce costs, methods of EVs func-tionalization (such as click chemistry, active loading using ultra-sound/extrusion/freeze-thaw cycles/electroporation/permeabilization) are being de-veloped, but they show side effects such as oxidative, destructive effects on molecules, or aggregation of EVs. Therefore, modified EVs obtained by any of the described methods must undergo quality control to evaluate morphology, integrity, homogeneous size, absence of EV aggregates, and absence of contaminants in the form of xenogenic components of the nutrient medium [112].

It is important to note that the size of the obtained EVs and the presence of a ten-dency to aggregation determine the possible route of administration of these vesicles. Furthermore, each batch of produced EVs must undergo verification of molecular composition and biological activity [112].

We believe that it is worth choosing a specific strategy for modifying vesicles based on their intended application. For example, giving priority to high encapsulation effi-ciency and applicability for most cargoes in the case of antitumor therapy. On the other hand, for regenerative medicine and immunomodulatory therapy, the most important criteria are minimal destructive effects on molecules, biocompatibility, and unchanged physicochemical parameters of EVs.» (Lines 564-580).

Comment

- Finally, section on challenges the community is facing in EV engineering need to be discussed. Critical comments on these will provide basis for new research and help accelerate the field.

Reply

As it was suggested we have added the section on Challenges and Future Directions in EV engineering (Please see Section 4, Lines 582-615, highlighted in blue).

Comment

- Minor comment: Figure 2 can be moved to page 3 or split into multiple figures and added at their respective sections (optional).

Reply

As it was suggested we have divided Figure 2 into different figures (Figures 2-7).